# The Corrosion Properties of Bronze Alloys in NaCl Solutions

**DOI:** 10.3390/ma16145144

**Published:** 2023-07-21

**Authors:** Zhiqiang Song, Ojiyed Tegus

**Affiliations:** 1Institute for the History of Science and Technology, Inner Mongolia Normal University, 81 Zhaowuda Road, Hohhot 010022, China; songzhiqiang@imnu.edu.cn; 2College of Physics and Electronic Information, Inner Mongolia Normal University, 81 Zhaowuda Road, Hohhot 010022, China

**Keywords:** bronze alloy, corrosion, NaCl solution, SEM, XRD, Tafel curve, impedance

## Abstract

Chloride ions play an important role in the corrosion of bronze through their active reactivity to copper alloys. The corrosion behavior of bronze alloys in NaCl solution was investigated by using X-ray diffraction (XRD), a scanning electron microscope (SEM), and electrochemical tests, with a special emphasis on the corrosion resistance of the α and δ phases in Cu-20 wt%Sn bronze alloys. The experimental results show that the corrosion current density of Cu-20 wt%Sn bronze alloys increases from 1.1 × 10^−7^ A/cm^2^ to 2.7 × 10^−6^ A/cm^2^ with the increase in the chloride ion concentration from 10^−3^ mol/L to 1 mol/L. After a soaking duration of 30 days, the matrix corrosion depth reaches 50 μm. The α phase of the alloys is easily corroded in NaCl solution, while the δ phase with high Sn content has strong corrosion resistance. This study provides relevant data for the analysis and protection of ancient bronze alloys.

## 1. Introduction

Bronze is mainly composed of elements such as copper and tin and exhibits excellent mechanical properties [1]. Although it is one of the earliest alloys used by humans, it still has widespread application in modern industry [2]. Prior to excavation, bronze artifacts unavoidably react with salts present in water or soil, causing oxidation–reduction reactions, hydrolysis reactions, and precipitation reactions between the bronze matrix and its corrosion products. Over a long period under earth, both the matrix and the surface of ancient bronzes undergo different degrees of corrosion [3].

Archaeologists have conducted extensive research on the corrosion and protection of bronzes [4,5,6,7,8,9,10,11]. There are some models that could explain the corrosion mechanism of bronze alloys. Some studies suggest that corrosion occurs preferentially in grains with impurities or inclusions [12], while others propose grain boundary corrosion as a prominent mode of degradation in bronze alloys [13]. Additionally, pitting corrosion is recognized as a common form of bronze corrosion [14,15]. When cooled from the liquid phase to the solid phase, Sn bronze alloys tend to segregate, forming Cu-rich α solid solution dendrites and a Sn-rich δ phase (Cu_41_Sn_11_). Despite the inherent corrosion resistance of bronze, selective corrosion of Cu-rich or Sn-rich phases results from alloy segregation [16]. Some studies indicate that the α phase in the alloy is preferentially corroded [17], while others report preferential corrosion of the δ phase [18]. Moreover, the corrosion state of bronze is closely related to the environmental conditions, such as humidity and various anions [19].

Chloride ions play a crucial role in the formation and transformation of bronze patina by increasing the solubility of the patina layer and enhancing the conductivity of the matrix [20]. Chloride ions readily penetrate defects and grain boundaries in the alloy matrix, leading to the formation of cuprous oxide and chlorides [21]. The initial formation of cuprous oxide occurs on the surface of the matrix, gradually passivating the bronze surface, while the presence of chloride ions promotes the transformation of the passivated layer into loosely packed copper chlorides [22]. The chloride ions can cause preferential corrosion of the Cu-rich phase in bronze alloys in high-oxygen seawater conditions, while the α + δ eutectic phase is susceptible to corrosion under low-oxygen conditions. Currently, there is extensive research on the corrosion behavior of copper alloys in seawater environments [23]. The occurrence and development of corrosion in bronze alloys are closely related to the presence of moisture and chloride ions, constituting a self-catalytic corrosion process [24]. When chloride ions are present, they react with the bronze matrix, forming a CuCl layer adjacent to the alloy matrix [25], and the surface morphology and elemental composition of corrosion products largely depend on the chemical composition of the alloy [26]. 

The above work mainly focuses on studying the corrosion laws of tin bronze alloys in different environments, most of which are about the effect of chloride ions on the corrosion of copper alloys. It has been proven that in chloride aqueous solutions, the passivation layer on copper alloys consists of copper oxides and copper chlorides [27]. However, there is relatively limited research on the influence of chloride ion concentration on the corrosion of copper-tin alloys, although higher chloride ion concentrations favor the occurrence of pitting corrosion [28]. Investigating the corrosion rate of bronze alloys in water environments with different chloride ion concentrations and the corrosion resistance of different phases within the alloy is needed for understanding the corrosion and protection of Cu-Sn alloys and bronzes.

This article reports the corrosion behavior of synthetic bronze alloys in NaCl solutions (1 mol/L, 0.1 mol/L, 0.01 mol/L, and 0.001 mol/L) [29] of different concentrations investigated by using XRD, SEM, the polarization curve method [30], and electrochemical impedance spectroscopy [31]. This article selects Cu-20 wt%Sn bronze for corrosion resistance research. The Cu-rich α phase and Sn-rich δ phase in Cu-20 wt%Sn bronze account for almost half each, making it more intuitive to compare the corrosion situation of the α phase and δ phase.

## 2. Materials and Methods

### 2.1. Materials

Copper and tin with 99.9% purity were used as raw materials. First, 100 g of copper was heated to 1200 °C in an induction furnace, and then 20 wt%Sn was added, at which point the temperature was kept constant for 5 min. In order to imitate the preparation process of ancient bronzes, the sample is prepared in air and naturally cooled. Finally, the ingot sample was cut into a 10 mm × 10 mm × 10 mm cube. Other materials used in the experiment included NaCl (from Fu Chen Chemical Reagents Co., Ltd., Tianjin, China), a AgCl electrode (from Shanghai Xianren Chemical Reagents Co., Ltd., Shanghai, China), a Pt electrode (self-made), and deionized water (self-made).

### 2.2. Methods

The measurements were conducted using a CHI660E electrochemical workstation (Shanghai Chenhua, Shanghai, China). Electrochemical impedance spectroscopy and polarization curve analysis were employed to investigate the effects of different immersion times and chloride ion concentrations on the corrosion of bronze alloys. Firstly, under stable open-circuit potential, electrochemical impedance spectroscopy was performed over a frequency range from 100 kHz to 0.1 Hz with a perturbation signal amplitude of 10 mV. Secondly, polarization curve testing was conducted at a scan rate of 20 mV/min within a scanning range of ±400 mV relative to the open-circuit potential. The sample’s structure was analyzed using XRD (PANalytical Empyrean, Almelo, The Netherlands). The metallic structures were observed and photographed under a metallographic microscope (ZEIZZ Observer A1m, Oberkochen, Germany). Scanning electron microscope (Hitachi TM3030, Tokyo, Japan) was used to observe the sample’s morphology, and EDS (Bruker Q70, Billerica, MA, USA) was used to analyze the elemental composition. 

## 3. Results and Discussion

### 3.1. Structure and Composition of the Bronze Alloys

Figure 1 presents the XRD patterns of Cu-20 wt%Sn bronze alloys without soaking and when soaked for 30 days in 0.1 mol/L NaCl solution. The analysis of the XRD patterns showed that the structure of the sample prior to soaking exhibited the coexistence of an α phase with a face-centered cubic structure and a δ phase with a complex cubic structure. In contrast to the sample soaked in NaCl solution, there were some diffraction peaks corresponding to the CuCl and Cu_2_O phases in addition to the α and δ phases.

Figure 2a shows the metallographic structure of the Cu-20 wt%Sn bronze. The morphology of the alloy indicates the presence of an α phase and an (α + δ) eutectic microstructure. It also can be seen that the sample contains defects and shrinkage voids.

Figure 2b displays the backscattered electrons (BSE) SEM image of the Cu-20 wt%Sn bronze. The BSE image contains information about the sample’s composition in the form of image contrast. The difference in contrast between the α and δ phases is clearly visible. The dark gray regions correspond to the α phase, while the light gray regions represent the δ phase.

Table 1 lists the compositions of the α and δ phases in the Cu-20 wt%Sn bronze. The main chemical composition of the sample consists of Cu and Sn elements, and the mass ratio of the elements in the sample is generally consistent with the alloy’s nominal composition. The α solid solution phase contains 13.50% Sn. The δ phase contains 23.82% Sn, which is slightly lower than the theoretical value of 32.6% in Cu_41_Sn_11_. The reason is that the actual measurement area for characteristic X-rays is larger than the selected area; therefore, it includes some of the α phase components.

### 3.2. Microscopic Corrosion Morphology and Products

To directly compare the corrosion resistance of the α and δ phases, we selected a Cu-20 wt%Sn bronze with a near-equal proportion of both phases for in-depth analysis. Figure 3 shows the SEM images of the sample surface soaked in NaCl solution for 30 days. Table 2 presents the EDS compositional analysis of the corroded sample. According to the results, the Cu-rich α-phase contains higher oxygen content than the Sn-rich δ-phase, and Cu is more depleted in the α-phase.

Figure 4 and Figure 5 respectively show the cross-sectional SEM image and the corresponding elemental distribution map of the Cu-20 wt%Sn bronze soaked in a 0.1 mol/L NaCl solution for 30 days. The cross-sectional SEM image reveals that the Cu-rich phase near the matrix surface is more susceptible to corrosion than the Sn-rich phase and that corrosion occurs along the grain boundaries. The element distribution map clearly shows severe copper depletion and an increase in oxygen content in the Cu-rich phase near the matrix surface. The surface (Figure 3) composition of the alloy soaked in NaCl solution for 30 days indicated higher oxygen content in the Cu-rich phase than in the Sn-rich phase, which is consistent with the elemental distribution along the cross-section (Figure 5). The corrosion depth near the matrix surface is approximately 50 μm. 

Sn is more active in NaCl solution than copper, so Sn is initially oxidized to form a protective Sn-rich passivation film on the surface, preventing further corrosion. Only when the passivation film is continuously eroded or breached does the copper start to corrode. Due to the Sn-rich δ-phase in the sample, the passivation film has strong protective properties and high corrosion resistance. According to Wang’s study [32], the corrosion product of tin is mainly SnO_2_, and the corrosion behavior of the alloy is influenced by the tin content. Alloys with higher tin content exhibit a lower corrosion current density and stronger corrosion resistance [33]. 

The elemental analysis results show that copper and chloride ions increase near the surface of the alloy, while in the alloy matrix, the Cu content of the α phase decreases. The Cu lost from the matrix and Cl ions in the solution form CuCl products on the surface of the sample. The XRD results (Figure 1) show the presence of Cu_2_O and CuCl phases in the post-immersion sample.

CuCl is an unstable intermediate product because it releases chloride ions during hydrolysis and oxidation processes [34], leading to the continuous dissolution of the bronze alloy matrix and the formation of Cu_2_O on the surface [35,36,37]. The Cu-rich α-phase in the matrix is more reactive with chloride ions to form CuCl. The combination of O_2_ generated after corrosion with the crevices and the Cu-rich phases in the matrix further contributes to the formation of metal oxides, resulting in accelerated corrosion of the Cu-rich phases in the alloy. Therefore, the copper loss is faster in the α-phase, while the δ-phase exhibits stronger corrosion resistance.

Figure 6 shows the SEM images and corresponding EDS spectra of Cu-20 wt%Sn during the initial corrosion stage (1 h) in different concentrations of NaCl solution. The composition of the samples was analyzed using EDS. The results indicate that the samples exhibited lower levels of oxygen (O) and chlorine (Cl) at the beginning of immersion in NaCl solutions of varying concentrations. As the solution concentration increased from 0.001 mol/L to 0.01 mol/L, the oxygen content in the samples increased. However, when the solution concentration increased from 0.01 mol/L to 1 mol/L, the oxygen content in the samples decreased. The chlorine content in the samples, on the other hand, increased with the increase in the solution concentration.

### 3.3. Electrochemical Corrosion Characteristics

Open circuit potential (OCP) is an important parameter for evaluating the corrosion behavior of alloys and is used to analyze the corrosion mechanism of bronzes. Figure 7 shows the OCP curves of the Cu-20 wt%Sn bronze in NaCl solutions of different concentrations. It can be seen that the concentration in NaCl solution has an impact on the corrosion potential of the bronze alloys once the self-corrosion potential reaches a stable state. With the increase in concentration, the self-corrosion potential of the sample increases and the OCP changes from −0.225 V to −0.02 V.

Figure 8 illustrates Tafel curves for the sample in NaCl solutions of different concentrations. The polarization curve was fitted using CHI660E software (CHI Version 14.05); the results are shown in Table 3. It can be seen that the corrosion current density of the sample increases with the increasing concentration of NaCl. The fitting results indicate that the corrosion current density of the bronze increases from 1.1 × 10^−7^ A/cm^2^ to 2.7 × 10^−6^ A/cm^2^ with a change in concentration from 10^−3^ mol/L to 1 mol/L. Through Stern’s formula *Rp* = *Ba* × *Bc*/[2.3 (*Ba* + *Bc*) × *I*_ccor_] (*Ba* and *Bc* are the Tafel slopes of the anode and cathode, respectively, and *Rp* is the polarization resistance). To calculate the polarization resistance *Rp* of the bronze alloys at different concentrations, *Rp* (10^−3^ mol/L) > *Rp* (10^−2^ mol/L) > *Rp* (10^−1^ mol/L) > *Rp* (1 mol/L). The higher the polarization resistance, the stronger the corrosion resistance of the alloy, indicating that bronze alloys have strong corrosion resistance in low-concentration NaCl solutions.

Figure 9a illustrates the Nyquist plot of the electrochemical impedance spectroscopy for the sample in NaCl solutions of different concentrations. In the Nyquist plot, the horizontal and vertical coordinates correspond to the real part (*Z*′) and imaginary part (*Z*″) of the complex transfer function. The capacitance arc of the alloy decreases with the increase in concentration. The size of the capacitance arc can be used to determine the charge transfer resistance and, consequently, the corrosion rate. Figure 9b shows the Bode plot, the horizontal coordinate represents the logarithmic frequency scale (Hz), and the vertical coordinate represents the phase shift of the transfer function of the system. From the Bode plot, it can be seen that the electrochemical impedance spectroscopy in 10^−3^, 10^−2^, 10^−1^, and 1 mol/L NaCl solutions exhibit only one capacitance arc, indicating the presence of a single time constant. This suggests that an oxide film does not form on the surface of the alloy during the initial 1 h immersion in NaCl solutions of different concentrations. The phase angle values for the different concentrations are 54.3, 53.5, 58.28, and 70.73 degrees. When the solution concentration is 10^−3^ and 10^−2^ mol/L, the change is not significant, and the concentration further increases, and the phase angle also increases.

The equivalent circuit diagram of the bronze in NaCl solution is shown in Figure 11a. In this circuit, Rs represents the resistance of the NaCl solution, R_ct_ corresponds to the charge transfer resistance, and Qdl is a constant phase element (CPE), which is the double-layer capacitance between the NaCl solution and the sample, as well as the non-ideal behavior of the double-layer capacitance. Table 4 provides the fitting results of the electrochemical components of the equivalent circuit for the Cu-20 wt%Sn bronze in NaCl solutions. The fitting results indicate that Rs decreases gradually with the increasing concentration. This leads to a decrease in resistance because the conductivity of the solution increases with the higher NaCl concentration. R_ct_ is an important indicator of corrosion rate. Higher charge transfer resistance indicates better corrosion resistance of the alloy. The fitting results show that within the range of 10^−3^ to 1 mol/L NaCl concentration, R_ct_ decreases with the increasing chloride ion concentration. The accelerated dissolution of Cu elements in the alloy with increasing solution concentration was confirmed by subsequent immersion experiments. This suggests that the increase in chloride ion concentration contributes to the corrosion rate of the bronze substrate.

Figure 10a shows the Nyquist plot; it demonstrates another set of parallel experiments where the Cu-20 wt%Sn bronze is soaked in a 0.1 mol/L NaCl solution for different immersion periods: 0 days, 3 days, and 30 days. Figure 10b shows the Bode plot of the sample; the phase angles at different times are 58.28, 52.58, and 22.37 degrees. As the soaking time increases, the phase angle becomes smaller; it shows only one time constant in the high-frequency region, indicating activation of the metal substrate and uniform corrosion on the electrode alloy surface without the formation of an oxide film. With an increase in the soaking time span to 3 days, the Bode plot of the alloy still exhibits only one time constant. However, when the soaking time is extended to 30 days, a double time constant appears in the Bode plot. The time constant in the high-frequency region corresponds to the response of the passive film interface, while the time constant in the medium–low-frequency region corresponds to the corrosion process of the metallic matrix. This suggests the presence of an oxide film on the electrode’s surface, which partially inhibits the corrosion process of the matrix. Figure 10c shows the Bode plot (magnitude) of the sample; as the soaking time increases, Z gradually decreases.

From the EIS tests and fitting results, it can be seen that the charge transfer resistance (R_ct_) decreases gradually with the increase in the chloride ion concentration in the NaCl solution. In chloride ion solutions, the Sn-rich δ-phase in the alloy demonstrates better corrosion resistance than the α-phase, indicating that the corrosion of the α-phase is the main factor affecting impedance changes. The increase in the solution concentration accelerates the dissolution of Cu elements. The results from the OCP, Tafel, and EIS tests are consistent and confirm that the increase in the chloride ion concentration contributes to the corrosion rate of the bronze matrix.

The impedance analysis from the soaking tests (Figure 10) reveals that the bronze initially undergoes uniform corrosion during immersion, and after 30 days, a protective film forms on the alloy surface. Based on the analysis of the experimental results mentioned above, the corrosion behavior and equivalent circuit diagram of the bronze in NaCl solution at different time intervals can be depicted, as shown in Figure 11. The schematic of the corrosion mechanism of the Cu-20 wt%Sn bronze after immersion in 0.1 mol/L NaCl solution is shown in Figure 12.

## 4. Conclusions

The present work aimed to describe the complex corrosion chemistry of a binary Cu-Sn alloy, understand the corrosion process of Cu-20 wt%Sn bronze in NaCl solution, and elaborate on the corrosion resistance of different phases in the alloy. From the experimental results, the main conclusions are as follows. Within the NaCl concentration range of 10^−3^ to 1 mol/L, the corrosion current density of the bronze increases from 1.1 × 10^−7^ A/cm^2^ to 2.7 × 10^−6^ A/cm^2^ as the chloride ion concentration increases, indicating that chloride ions accelerate corrosion of the bronze. In the immersion experiment with a 0.1% NaCl solution, after 30 days of immersion, significant copper loss occurs in the α-phase in the alloy, resulting in the formation of a film composed of CuCl and Cu_2_O on the surface, with a maximum depth of 50 µm. The Sn-rich δ-phase in the bronze exhibits stronger corrosion resistance in chloride ion solutions.

## Figures and Tables

**Figure 1 materials-16-05144-f001:**
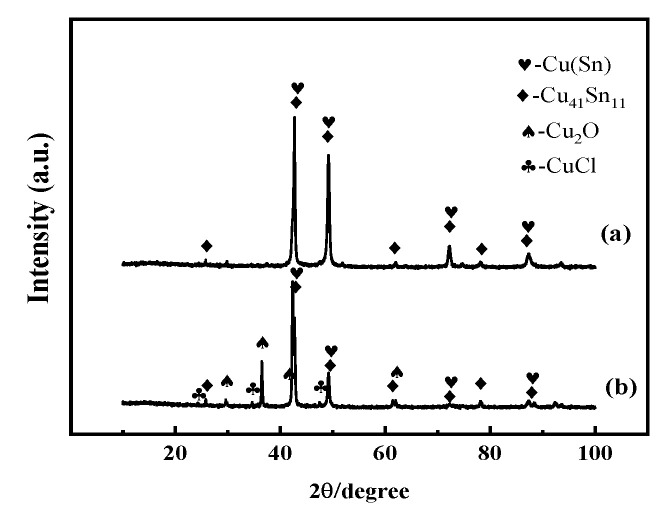
XRD patterns of the Cu-20 wt%Sn bronze in 0.1 mol/L NaCl solution. (**a**) Without soaking and (**b**) when soaked for 30 days.

**Figure 2 materials-16-05144-f002:**

(**a**) Optical micrograph and (**b**) SEM image of the Cu-20 wt%Sn bronze (the red circle area indicated by the arrow is α or δ phase).

**Figure 3 materials-16-05144-f003:**
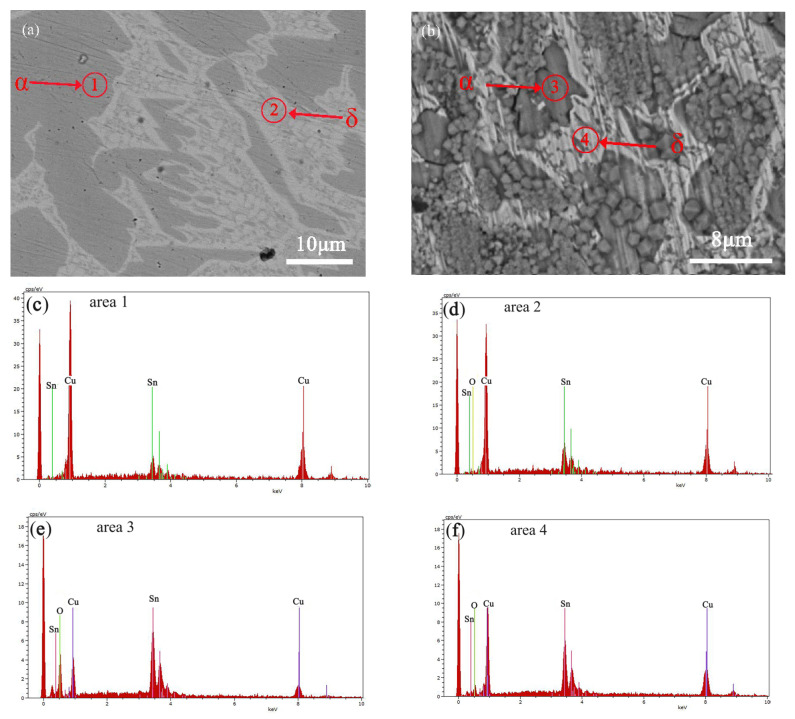
SEM-EDS image of the Cu-20 wt%Sn bronze soaked in 0.1 mol/L NaCl solution, without corrosion (**a**,**c**,**d**) and with corrosion 30 days (**b**,**e**,**f**), the red circle area indicated by the arrow is α or δ phase.

**Figure 4 materials-16-05144-f004:**
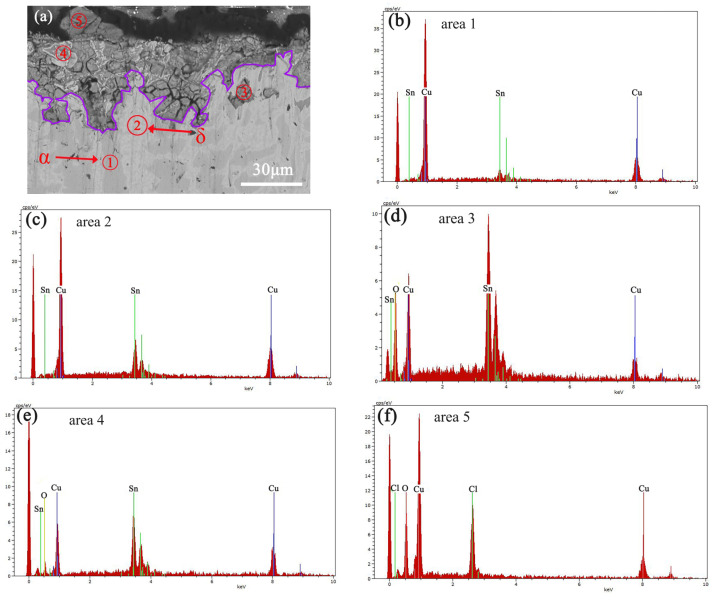
SEM-EDS image of the cross-section of the Cu-20 wt%Sn bronze soaked in 0.1 mol/L NaCl solution for 30 days: SEM (**a**) and EDS (**b**–**f**). Areas (1–5) represent different areas of the sample, arrows indicate α or δ phase, the area above the purple line shows the corrosion area.

**Figure 5 materials-16-05144-f005:**
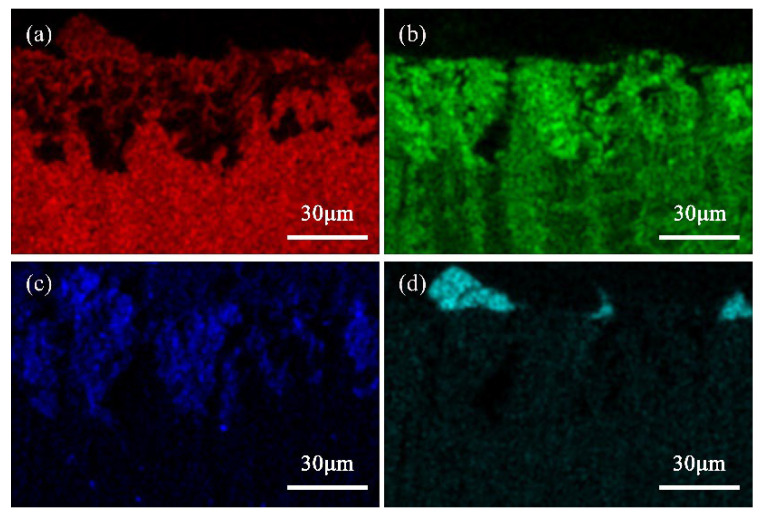
Element distribution in the cross-section of the Cu-20 wt%Sn bronze soaked in 0.1 mol/L NaCl solution for 30 days: (**a**) Cu, (**b**) Sn, (**c**) O, and (**d**) Cl.

**Figure 6 materials-16-05144-f006:**
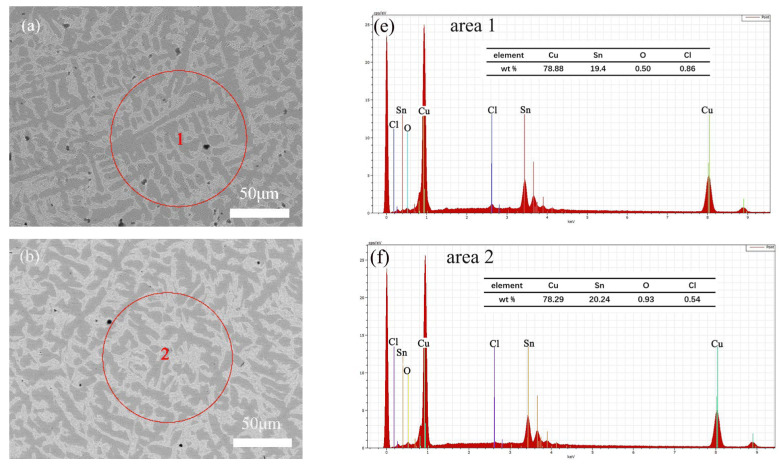
SEM images and EDS spectra of Cu-20 wt%Sn in NaCl solution (1 h): (**a**,**e**) for 1 mol/L, (**b**,**f**) for 0.1 mol/L, (**c**,**g**) for 0.01 mol/L, and (**d**,**h**) for 0.001 mol/L.

**Figure 7 materials-16-05144-f007:**
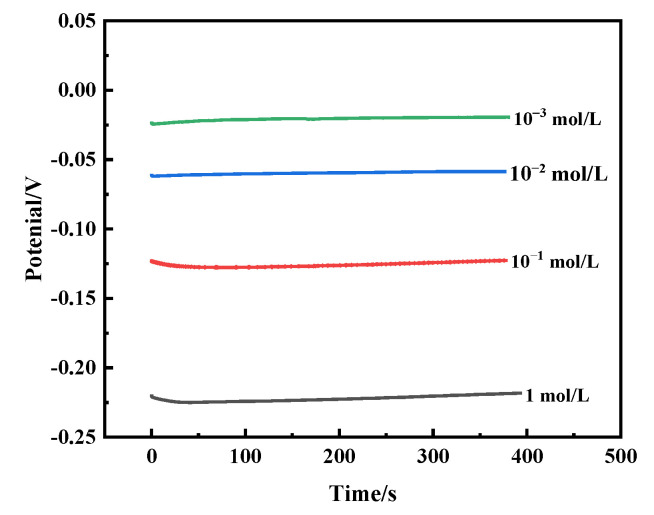
OCP curves of the Cu-20 wt%Sn bronze in NaCl solutions.

**Figure 8 materials-16-05144-f008:**
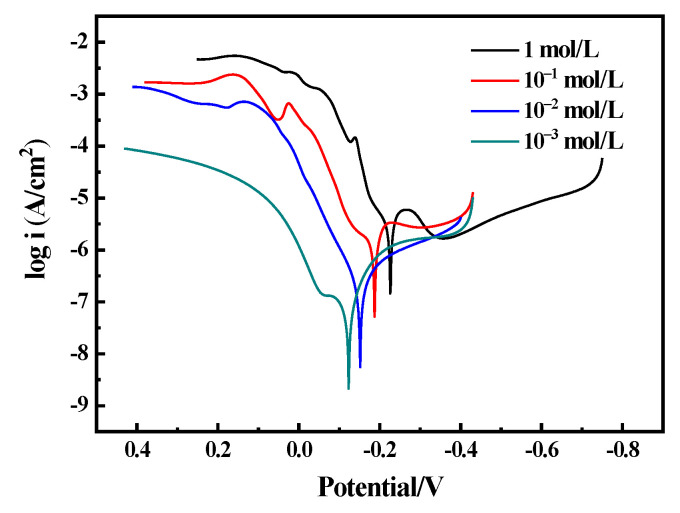
Tafel curves of the Cu-20 wt%Sn bronze in NaCl solutions.

**Figure 9 materials-16-05144-f009:**
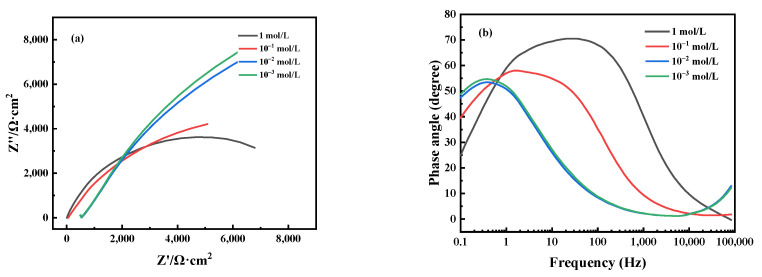
(**a**) Nyquist plot and (**b**) Bode plot of the Cu-20 wt%Sn bronze in NaCl solutions.

**Figure 10 materials-16-05144-f010:**
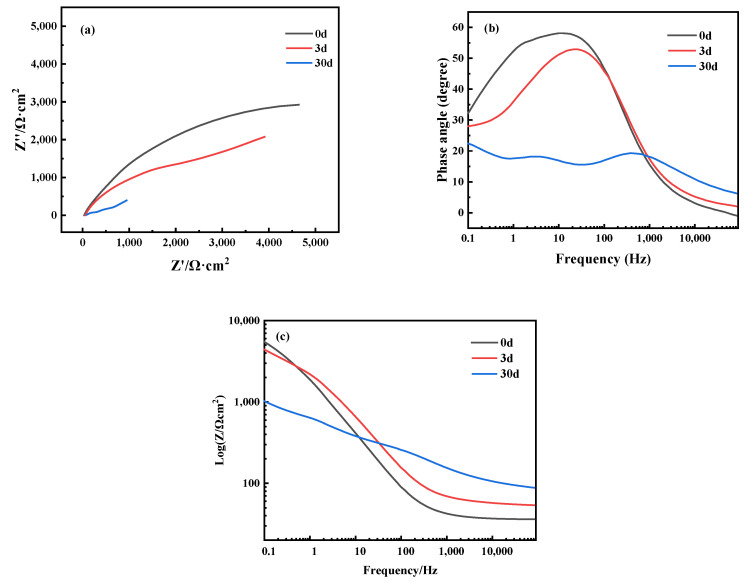
(**a**) Nyquist plot and (**b**,**c**) Bode plots of the Cu-20 wt%Sn bronze soaked in 0.1 mol/L NaCl solution for different time spans.

**Figure 11 materials-16-05144-f011:**
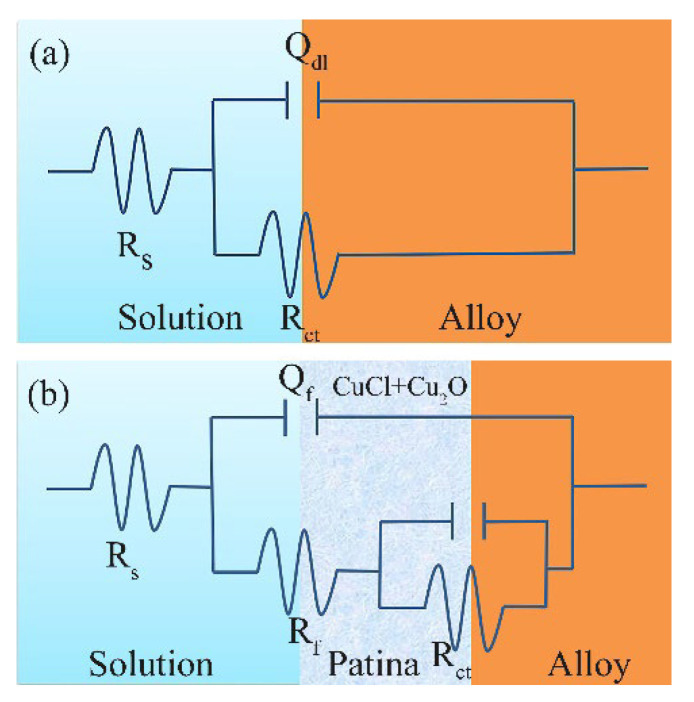
Equivalent circuits show the Cu-20 wt%Sn bronze soaked in 0.1 mol/L NaCl solution initially (**a**) and for 30 days (**b**).

**Figure 12 materials-16-05144-f012:**
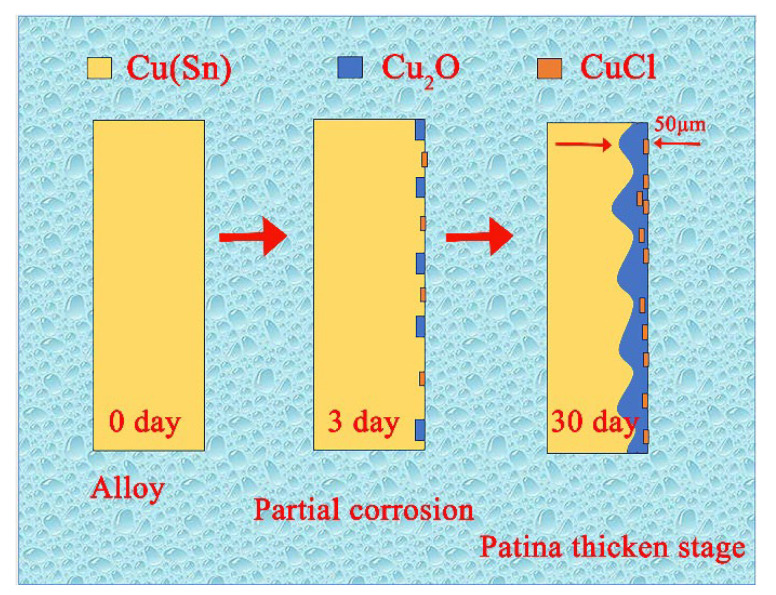
Schematic of the corrosion mechanism of the Cu-20 wt%Sn bronze after immersion in 0.1 mol/L NaCl solution (the arrow direction indicates that the soaking time of the sample increases sequentially).

**Table 1 materials-16-05144-t001:** The compositions of different phases in the Cu-20 wt%Sn bronze.

Position	Sn (wt%)	Cu (wt%)
average value	20.12	79.88
α-phase	13.50	86.50
(α + δ) phase	23.82	76.18

**Table 2 materials-16-05144-t002:** EDS analysis of the Cu-20 wt%Sn bronze with and without corrosion.

Position		Sn (wt%)	Cu (wt%)	O (wt%)
Without corrosion	α-phase	13.50	86.50	<D.L.
	δ-phase	23.82	76.18	<D.L.
With corrosion	α-phase	48.41	29.53	22.06
	δ-phase	37.32	59.52	3.16

**Table 3 materials-16-05144-t003:** Fitting parameters of the polarization curves of the Cu-20 wt%Sn bronze in NaCl solutions.

Solution	*E*_corr_/V	*I*_corr_/A·cm^2^	*Ba*/V^−1^	*Ba*/V^−1^	*Rp*/KΩ
1 mol/L	−0.223	2.769 × 10^−6^	8.801	10.913	7.9648
10^−1^ mol/L	−0.187	4.578 × 10^−7^	2.745	11.160	68.301
10^−2^ mol/L	−0.152	3.032 × 10^−7^	3.921	13.843	80.723
10^−3^ mol/L	−0.123	1.101 × 10^−7^	31.944	9.348	95.635

**Table 4 materials-16-05144-t004:** EIS fitting results of the Cu-20 wt%Sn bronze in NaCl solutions.

Solution	Rs/(Ω·cm^2^)	Qdl/(Ω^−1^·cm^2^·s ^n^)	n	R_ct_/(Ω·cm^2^)
1 mol/L	6.44	6.148 × 10^−5^	0.8185	9406
10^−1^ mol/L	54.92	7.777 × 10^−5^	0.7691	16,306
10^−2^ mol/L	478.5	1.333 × 10^−4^	0.6672	37,271
10^−3^ mol/L	58,556	1.168 × 10^−4^	0.6918	58,556

## Data Availability

The data that support the findings of this study are available from the corresponding author upon request.

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
