# Peer review of "The Corrosion Properties of Bronze Alloys in NaCl Solutions"

_materials, 2023, doi:10.3390/ma16145144_

Round 1

Reviewer 1 Report

The manuscript n. Materials-2505242 titled “Corrosion Properties of Bronze Alloy in NaCl Solutions” achieves the purpose defined in the abstract, it is well structured, it is scientifically robust and technically sound, and it is written with care.

I’m convinced that the Research is not innovative, but both discussion and conclusions allow to increase the knowledge of studied field.

In my opinion, the manuscript is of a very good standard. Therefore, I believe that the manuscript can be accepted for publication after the  following Minor Revisions:

Abstract. Replace “10-4mol/L to 10-1mol/L” with “10-4 mol/L to 10-1 mol/L”.

Keywords. The authors should add SEM and XRD.

Pag. 1. The authors write “Archaeologists have conducted extensive research on the corrosion and protection of bronzes [4-9]”. The sentence is correct, but the authors should add other references such as:

D.A. Scott, Copper and bronze in art: corrosion, colorants, conservation. The Getty Conservation Institute, Los Angeles, (2002) 515.

Buccolieri, G.; Buccolieri, A.; Donati, P.; Marabelli, M.; Castellano, A..; Portable EDXRF investigation of the patinas on the Riace Bronzes. Nuclear Ins truments and Methods in Physics Research, Section B: Beam Interactions with Materials and Atoms 2015, 343,101-109. DOI: 10.1016/j.nimb.2014.11.064.

Introduction. The authors write “….polarization curve method….”. The authors must add some Reference.

Introduction. The authors write “……electrochemical impedance spectroscopy.”. The authors must add some Reference.

Caption of Figure 1. Replace “0.1mol/L NaCl” with “0.1 mol/L NaCl”.

Figure 2. The authors must increase the quality of size bar.

Table 1. Replace “Sn (wt.%)” with “Sn (wt %)”.

Table 1. Replace “Cu (wt.%)” with “Cu (wt %)”.

Figure 3. The authors must increase the quality of size bar.

Table 2. The authors must indicate Table 2 in the manuscript.

Table 2. Replace “Sn (wt.%)” with “Sn (wt %)”.

Table 2. Replace “Cu (wt.%)” with “Cu (wt %)”.

Table 2. Replace “O (wt.%)” with “O (wt %)”.

Table 2. The authors write “zero” for the oxygen concentration. The authors must report ”<D.L.“. Logically, the authors must calculate the limit of detection.

Figure 4. The authors must increase the quality of size bar.

Caption of Figure 4. Replace “0.1mol/L NaCl solution” with “0.1 mol/L NaCl solution”.

Caption of Figure 9. Replace “0.1mol/L NaCl” with “0.1 mol/L NaCl”.

Caption of Figure 10. Replace “0.1mol/L NaCl” with “0.1 mol/L NaCl”.

Conclusions. Replace “10-4 to 10-1” with “10-4 to 10-1”.

Conclusions. The authors should write the conclusions without dividing the text into three points.

Reviewer 2 Report

materials-2505242

1.       The abstract should start with the most significant subject of the paper. The reader should understand the main topic of the paper from the first sentence. In addition, you need to focus more on quantitative information, not qualitative ones.

2.       All abbreviations such as XRD, SEM should be defined for the first time in the abstract.

3.       Authors should identify gaps in existing studies for potential future research. The authors did not explain the novelty and significance of their work in the introduction section.

4.       More information regarding material composition must be given. You have to clarify the reasons for selecting material composition Cu containing 20 wt%Sn? You have to add a reference for that or explain how you got this composition as optimal ones. In addition, the author investigated the corrosion behavior including polarization test and electrochemical impedance spectroscopy of the sample for the sample in NaCl solutions with different concentrations. You have to clarify the reasons for measuring the corrosion rate and corrosion resistance of the Cu containing 20 wt%Sn in NaCl solutions with different concentrations 1mol/L, 0.1mol/L, 0.01mol/L and 0.001mol/L. You have to add a reference for that or explain how you got these concentrations 1mol/L, 0.1mol/L, 0.01mol/L and 0.001mol/L as optimal ones.

5.       The authors indicate the EDS analysis of cu-20 wt%sn bronze with and without corrosion. However, the EDS spectra of each point or area is not presented.

6.       Element distribution in the cross-section of Cu-20 wt%Sn bronze soaked in 0.1mol/L NaCl solution for 30 days require scale bar and scale value. In addition, the EDS spectra of the sample should be presented.

7.       I suggest that the author present the surface morphology of the sample after immersion in NaCl solutions with different concentrations 1mol/L, 0.1mol/L, 0.01mol/L and 0.001mol/L.

8.       This paper suffers from presentation of poor corrosion data, thus I suggest that the author present the XRD analysis and/or FTIR analysis the Cu-20 wt%Sn sample after immersion in NaCl solutions with different concentrations 1mol/L, 0.1mol/L, 0.01mol/L and 0.001mol/L.

9.       I suggest that the author present the schematic of the corrosion mechanism of the Cu-20 wt%Sn sample after immersion in NaCl solutions

10.   Some references about Cu-based alloy corrosion behavior may be useful for this review article: Transactions of Nonferrous Metals Society of China 25 (2015) 1158-1170.  In addition, surprisingly small references to the materials in the literature despite the large relevant literature there. This should be improved. There are several important papers in recent literature.   

11.   Regarding Fig. 8 and 9, the value of phase angle should be stated in the main text.

12.   The author only present Bode phase plot, however, Bode magnitude plot should present beside Bode phase plot Cu-20 wt%Sn sample after immersion in NaCl solutions, and the value of impedance modulus at a low-frequency (|Z|) should be stated in the main text.

13.   I suggest that you include a summary of the corrosion resistance properties derived from the polarization curves such as: the anodic and cathodic Tafel constants (Ba and Bc, respectively) determined by application of the ButlerVolmer equation where EI=0 is the corrosion potential. The fitting was performed according to the non-linear least square fit method of Levenberg-Marquardt, which provides a numerical solution to the problem of minimizing a nonlinear function over a space of parameters. From the Stern and Geary relationship, the polarization resistance (Rp) can be evaluated. The values of βa, βc and Icorr indicated in Table 3 must correspond to given anodic and cathodic potential limits also indicated in this table 3. You must also include the relationship that permits you to evaluate the corrosion rate (CR). 

Reviewer 3 Report

This work describes the behavior of bronze alloy in NaCl solution, which is an important issue in the field of corrosion. The authors used standard techniques of XRD analysis, SEM electrochemical tests. The authors correctly presented the results of the initial sample and the samples exposed to corrosion. The results are adequately discussed and are consistent with previous research.

I recommend that the paper be published with minor updates

1. Please check the references (special reference [12])

2. cuprous oxide to replace with copper oxide, copper was used throughout the paper (sentence following reference [18])

3. In Figures 6, 7, 8 and in Tables 3 and 4, the authors write that the concentrations of NaCl were 1mol/L, 0.01 mol/L, 0.001mol/L, 0.0001 mol/L... and in the text it says that the concentrations were from 10-1mol/L to 10-4 mol/L ... please correct this

4. In the text 10-4 mol/l  should not be 10-4 mol/l

Round 2

Reviewer 2 Report

Authors have done the part of corrections and suggestions. However, some minor corrections and explanation must be done previous to publish the paper in the Materials. Specific questions and comments are listed as follow

Point 1: I suggest that the author present the surface morphology of the sample after immersion in NaCl solutions with different concentrations 1mol/L, 0.1mol/L, 0.01mol/L and 0.001mol/L. The author present the image but they did not transfer to the main text. the surface morphology of the sample after immersion in NaCl solutions with different concentrations 1mol/L, 0.1mol/L, 0.01mol/L and 0.001mol/L should be shifted from the reviewer response to the main text.

Author Response

Point 1: I suggest that the author present the surface morphology of the sample after immersion in NaCl solutions with different concentrations 1mol/L, 0.1mol/L, 0.01mol/L and 0.001mol/L. The author present the image but they did not transfer to the main text. the surface morphology of the sample after immersion in NaCl solutions with different concentrations 1mol/L, 0.1mol/L, 0.01mol/L and 0.001mol/L should be shifted from the reviewer response to the main text.

Response 1: Thank you for your suggestion. We have added the images as requested and marked them in the manuscript. The image results are described in the main text(Figure 6).